# Optimizing cardiovascular risk assessment and registration in a developing cardiovascular learning health care system: Women benefit most

T. Katrien J. Groenhof[1], Saskia Haitjema[2], A. Titia Lely[3], Diederick E. Grobbee[1], Folkert W. Asselbergs[4,5,6], Michiel L. Bots[1]*, on behalf of the UCC-CVRM and UPOD Study groups[¶]

1 Julius Center for Health Sciences and Primary Care, University Medical Center Utrecht, Utrecht University, Utrecht, The Netherlands, 2 Laboratory of Clinical Chemistry and Haematology, University Medical Center Utrecht, Utrecht University, The Netherlands, 3 Wilhelmina Children's Hospital Birth Centre, University Medical Center Utrecht, Utrecht, The Netherlands, 4 Department of Cardiology, Division Heart & Lungs, University Medical Center Utrecht, Utrecht University, The Netherlands, 5 Institute of Cardiovascular Science, Faculty of Population Health Sciences, University College London, London, United Kingdom, 6 Health Data Research UK, Institute of Health Informatics, University College London, London, United Kingdom

¶ Membership of the UCC-CVRM and UPOD Study groups are listed in the Acknowledgments.
* m.l.bots@umcutrecht.nl

**Data Availability Statement:** The data are based on information from confidential electronic health records. As such data cannot be made publicly

## Abstract

Since 2015 we organized a uniform, structured collection of a fixed set of cardiovascular risk factors according the (inter)national guidelines on cardiovascular risk management. We evaluated the current state of a developing cardiovascular towards learning healthcare system–the Utrecht Cardiovascular Cohort Cardiovascular Risk Management (UCC-CVRM)—and its potential effect on guideline adherence in cardiovascular risk management. We conducted a before-after study comparing data from patients included in UCC-CVRM (2015–2018) and patients treated in our center before UCC-CVRM (2013–2015) who would have been eligible for UCC-CVRM using the Utrecht Patient Oriented Database (UPOD). Proportions of cardiovascular risk factor measurement before and after UCC-CVRM initiation were compared, as were proportions of patients that required (change of) blood pressure, lipid, or blood glucose lowering treatment. We estimated the likelihood to miss patients with hypertension, dyslipidemia, and elevated HbA1c before UCC-CVRM for the whole cohort and stratified for sex. In the present study, patients included up to October 2018 (n = 1904) were matched with 7195 UPOD patients with similar age, sex, department of referral and diagnose description. Completeness of risk factor measurement increased, ranging from 0%-77% before to 82%-94% after UCC-CVRM initiation. Before UCC-CVRM, we found more unmeasured risk factors in women compared to men. This sex-gap resolved in UCC-CVRM. The likelihood to miss hypertension, dyslipidemia, and elevated HbA1c was reduced by 67%, 75% and 90%, respectively, after UCC-CVRM initiation. A finding more pronounced in women compared to men. In conclusion, a systematic registration of the cardiovascular risk profile substantially improves guideline adherent assessment and decreases the risk of missing patients with elevated levels with an indication for treatment. The sex-gap

available. Requests for use of the data can be directed to the UCC-CVRM project office (email: ucc@umcutrecht.nl). The request will be taken to the steering committee of UCC-CVRM and discussed further taking Dutch privacy and legal regulations into account. This approach has been detailed in the rationale and design publication of the UCC initiative. Eur J prev Cardiol 2017 by Asselbergs FW et al. (PMID: 28128643).

**Funding:** The UCC-CVRM is primarily financed by the UMC Utrecht as it is care as usual. UCC-CVRM as a whole, and via MLB as chair of the UCC-CVRM steering committee, was partly supported by a grant from the Netherlands Organization for Health Research and Development (#8480-34001) to develop feedback procedures, not for salary. The funders had no role in study design, data collection and analysis, decision to publish, or preparation of the manuscript.

**Competing interests:** The authors have declared that no competing interests exist.

disappeared after UCC-CVRM initiation. Thus, an LHS approach contributes to a more inclusive insight into quality of care and prevention of cardiovascular disease (progression).

## Author summary

The (inter)national guidelines for management of cardiovascular risk state that for all patients who come to a health care provider for the evaluation of their cardiovascular risk or symptoms a certain set of cardiovascular risk factors should be measured. In order to facilitate this recommendation across all hospital specialisms that take care of patients who come for the evaluation of their cardiovascular risk or symptoms, we organized in our hospital a uniform, structured collection of a fixed set of cardiovascular risk factors (the UCC-CVRM initiative) as part of a learning health care system (LHS). In the present report, we evaluated the effect of this approach on adherence to the guideline by comparing the guideline adherence before and after UCC-CVRM. We found that the adherence to the guideline improved considerably, more pronounced in women than in men. Furthermore, UCC-CVRM lead to a reduction of the risk of missing patients with elevated levels with an indication for treatment. We conclude that such an LHS approach contributes to a more inclusive insight into quality of care and prevention of cardiovascular disease (progression).

## Introduction

Healthcare is challenged by a growth in patient numbers. Furthermore, these patients are of higher age, suffer from multiple diseases–including a higher portion of chronic diseases-, and use more medications simultaneously [1]. Traditionally, evidence that forms the basis of care is derived from trials and cohorts. Yet, insights from trials are not always compatible with routine care. This science to care gap can in part be explained by differences in patient characteristics due to strict selection criteria or differences in care setting [2]. For example women are underrepresented in cardiovascular studies and have shown to receive substandard care in terms of risk management [3,4]. Altogether, the science to care gap results in modest use of evidence based findings in clinical practice, increased off label treatment, and non-adherence to guidelines. In return, the potential of routine care data is not captured fully, potentially resulting in a loss of valuable information on the course of disease and treatment in real life. This sparked the interest for a learning healthcare system (LHS).

The LHS was first described in 2007 as a system where routine care delivery and evidence synthesis are connected in a cycle [5]. In a LHS, routine clinical data is used to evaluate quality and safety of care and to generate evidence and drive discoveries, that in turn can be implemented back into routine care when using the LHS cycle (Fig 1). Because the cycle starts at routine care and uses data that has already been collected for care purposes, learning health care systems have the potential to generate evidence more efficient in terms of quality, time and costs [1]. Transformation from a traditional healthcare system to such a cyclic LHS approach requires efforts from all stakeholders in care and science, starting at generating awareness and adoption of evidence based medicine and generating sufficient quality input data [6]. Maturation of a LHS involves a shift from a more passive disconnected evaluation of care, to full integration of research into routine care [6]. Since 2007, many initiatives around the LHS arose, yet remained mostly described in theory [7]. A great contrast with the essence of the LHS theorem: "learning by doing" [5].

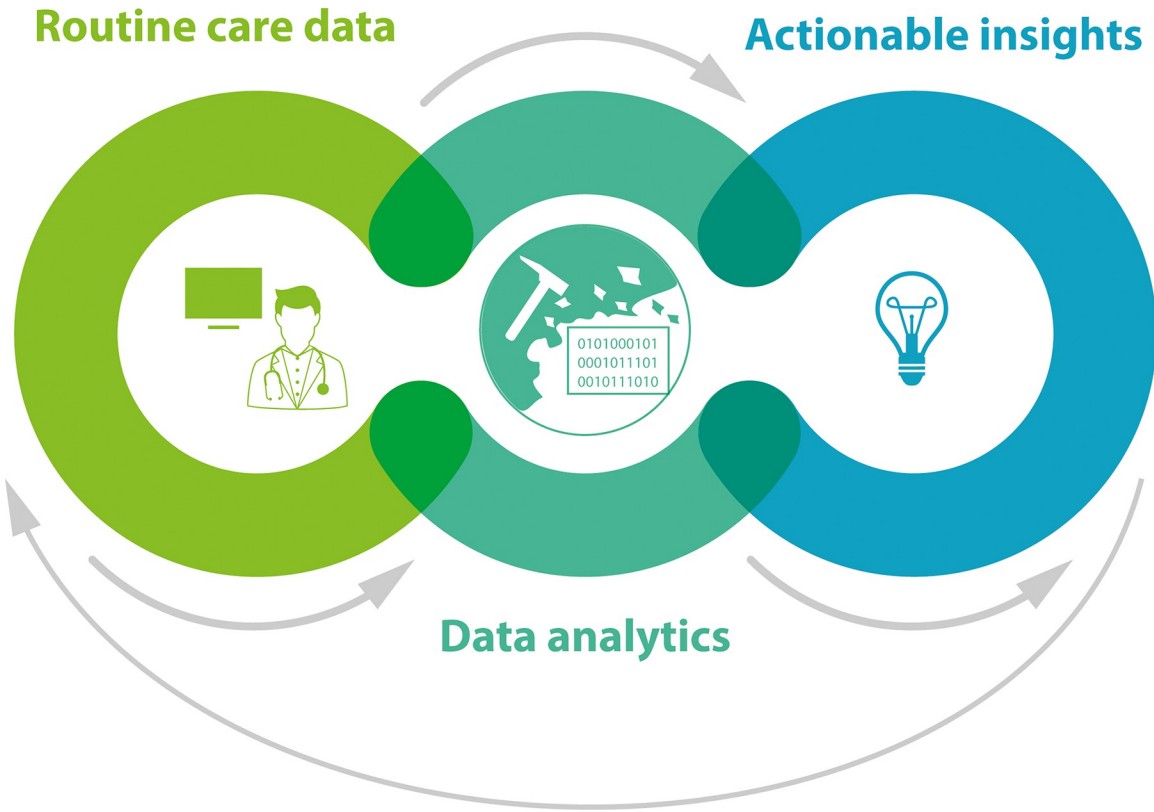

**Fig 1. Learning healthcare system cycle.**

In 2016, the Center for Circulatory health of the University Medical Centre Utrecht, Utrecht, The Netherlands, started the Utrecht Cardiovascular Cohort Cardiovascular Risk Management (UCC-CVRM) initiative: a cardiovascular LHS [8]. Cardiovascular disease (CVD) (risk) management is notoriously difficult because of multi-morbidity and different phenotypes and severities of disease. Guidelines provide support and advocate cardiovascular risk factor management in all cardiovascular patients. Still, prior research has shown differences in completeness of CVRM measurement between clinical specialties and patient groups: a missed opportunity prevent disease [8,9]. In the cardiovascular field, other attempts for LHS are described by Maddox et al [1] in the domains of "science and informatics", "patient-clinician partnerships", "incentives", and "development of a continuous learning culture". Most initiatives focus on either of the domains, using for example EHR data for predictive modelling [amarasingham et al. med care 2010; 48:981–988], using EHR data for effective cohort identification or eCRF extraction [33248277], identifying incentives that are associated with clinical improvement, using the LHS as a culture shift for continuous learning [1]. The UCC-CVRM created an infrastructure for uniform, structured collection of cardiovascular risk profile in routine clinical care of all departments treating patients with cardiovascular disease(s). This enables to provide feedback to physicians on quality of care, and generate new evidence on etiology, diagnostics, prognosis, and therapy, including efficacy, safety and cost-effectiveness for cardiovascular disease (risk). Thus, UCC was designed to be able to facilitate all domains, catalyzed by the basis of solid cardiovascular risk assessment. Furthermore, we included all patients visiting any department with a link to/dedicated to cardiovascular disease, connecting

8 departments in our hospital (and primary care) and thus creating a synergetic basis for interdisciplinary collaborations.

Here we evaluated the current state of this developing cardiovascular learning healthcare system and it potential effect on guideline adherence in cardiovascular risk management.

## Methods

### Study design and population

All data from the EHRs of the UMC Utrecht is extracted to the Utrecht Patient Oriented Database (UPOD). In short, UPOD comprises all digital traces of the UMC Utrecht, e.q. clinical information, demographic data, medication, diagnoses and laboratory measurements. UPOD extracted this data from 2003 onwards, encompassing data of more than 2.3 million individual patients to date [10,11]. A complete description of the UPOD database has been published elsewhere [10].

In May 2015, the first patient was included in the Utrecht Cardiovascular Cohort (UCC-CVRM) in the University Medical Center Utrecht. The UCC-CVRM is a prospective cohort study targeted to uniform assessment and registration of the guideline based cardiovascular risk profile in all patients presenting with a (risk factor for) cardiovascular disease within routine care. A detailed description of the protocol is published elsewhere. All data collected in UCC-CVRM is also captured in the EHR of the UMC Utrecht at the uniform structured locations.

We conducted a before-after study comparing data from patients included in UCC-CVRM and patients treated in the period before UCC-CVRM initiation who would have been eligible for UCC-CVRM from UPOD. We used data from patients visiting outpatient clinics who provided a written informed consent for UCC-CVRM from January 2016 up to December 31th 2018, and UPOD patients from January 1, 2013 to December 31, 2015. UPOD patients were matched to UCC-CVRM patients 1:4 based on age, sex, department of referral and inclusion in UCC-CVRM and diagnosis at inclusion. Diagnosis at inclusion was defined as the diagnosis treatment code registered closest to date of inclusion in UCC-CVRM. For UPOD, a theoretical inclusion date was defined by taking the time difference between inclusion and diagnosis for the UCC-CVRM patient date and applying this difference to the date of diagnose code for the matched UPOD patient. Throughout the manuscript, we will refer to the UPOD population as the "before UCC-CVRM" patients.

### Ethics and privacy

UCC-CVRM has been approved by the Institutional Review Board of the UMC Utrecht and all data is handled according to privacy regulations [2]. UPOD, encompassing secondary use of EHR data, is in accordance with Institutional Review Board (IRB) and privacy regulations of the UMC Utrecht: clinical data can be used for scientific purposes if patients cannot be identified directly from the data. All patients are informed through the opt-out procedure, a general UMC Utrecht procedure via which patients can object to use of their clinical data for scientific evaluations.

### Data collection

We collected data on sex, age, cardiovascular history and lifestyle factors, physical measurements (BMI and blood pressure), laboratory measurements including total cholesterol, high-density lipoprotein-c (HDL-c), low-density lipoprotein-c (LDL-c), triglycerides, estimated glomerular filtration rate (eGFR) according to the CKD-EPI formula, hemoglobin, and glycated

**Table 1. Data sources and time window per variables stratified for study population.**

|  | UCC-CVRM | Before UCC-CVRM |  |
| --- | --- | --- | --- |
| **Variables** | **Data source** | **Data source** | **Time window** |
| Demographics |  |  |  |
| Age | Structured questionnaire | General hospital administration |  |
| Sex | Structured questionnaire | General hospital administration |  |
| Cardiovascular history |  |  |  |
| CVD events | Structured questionnaire | Diagnosis and financial billing codes | Before inclusion date* |
| Risk factor history | Structured questionnaire | Diagnosis and financial billing codes | Inclusion date to 42 days after |
|  |  | Prescription of medication for that risk factor |  |
| Lifestyle |  |  |  |
| Smoking | Structured questionnaire | Text mining algorithm | 365 before to 7 days after |
|  |  | (Un)Structured questionnaires Free text |  |
| Physical activity | Structured questionnaire | Not extractable | Inclusion data to 180 days after N/a |
| Physical measurements |  |  |  |
| BMI | Structured measurements field EHR | Structured measurements field EHR |  |
| Blood pressure | Structured measurements field EHR | Structured measurements field EHR | Inclusion date to 365 days after |
| Laboratory measurements |  |  |  |
| Hb, total cholesterol, HDL-c, LDL-c, triglycerides, eGFR (CKD EPI formula), creatinine, HbA1c | Laboratory measurements | Laboratory measurements | 60 days before to 60 days after inclusion date |
| Cardiovascular medication use | Structured questionnaire | Extracted form electronic prescription system using the ATC codes starting with A10, B01, B02A, and C02 to C10. | 90 days before to 90 days after |

* the date of the matching diagnose code was used as a theoretical inclusion date

hemoglobin (HbA1c) [12]. An overview of variables and data sources is listed in Table 1. We predefined time windows between inclusion data and variable measurement for data extraction per variable (Table 1).

## Outcome definitions

First, we compared the proportion of measured risk factors before UCC-CVRM and after UCC-CVRM in the full cohort and stratified for sex. We also evaluated factors (UCC-CVRM, sex, interaction of UCC-CVRM and sex, age, history of CVD, history of diabetes) associated with measurement of smoking, SBP, LDL-c, eGFR and HbA1c. Second, we assessed if the patients had an indication for (start or change) of treatment for elevated blood pressure, LDL-c, and HbA1c [13]. The blood pressure target was <140/90mmHg, the LDL-c target was <2.5 mmol/L (at the time the 1.8mmol/L target for the very high risk population had not been adopted yet in The Netherlands), and the HbA1c target was <53mmol/L for patients with type II diabetes. Third, we calculated the proportion of missed uncontrolled hypertension, dyslipidemia, and HbA1c by multiplying the proportion of patients that were not measured by the proportion of patients with an off target measurement. Then, we calculated the likelihood to miss patients with uncontrolled hypertension, dyslipidemia, and elevated HbA1c before

UCC-CVRM by dividing the proportions missed before and after UCC-CVRM initiation. Fourth, we evaluated time trends in consent consults as a starting point for data collection. Finally, we provided feedback on clinical management of the most extreme measurements to evaluate awareness of guideline adherent CVRM. The most extreme values were defined as: a blood pressure above 180/110mmHg, a total cholesterol >8 mmol/L, a HbA1c level above 48mmol/mol without reported diabetes, an eGFR <30ml/min, and alcohol use >10 units per week [12].

## Statistical analysis

Statistical analyses were conducted in R studio (version 3.4.1, Copyright (C) 2017 The R Foundation for Statistical Computing). We used student t- tests to compare normally distributed continuous variables and Pearson chi square or fisher's exact test where appropriate for proportions. Associations of measurement of risk factors with UCC-CVRM initiation, sex, interaction of UCC-CVRM initation and sex, age, history of CVD, history of diabetes) were analyzed using generalized linear models with binomial family and logit link (logistic regression).

## Results

### Patient selection and characteristics

Up to December 31st 2018, 1904 out-patient clinic patients were included in UCC-CVRM. We could match these patients to 7195 patients in UPOD (1:3.8 match). Baseline characteristics of both populations are described in S1 Table. Prevalence of current smoking was higher before UCC-CVRM (23%) versus after UCC-CVRM initiation (12%). Other risk factors showed similar distributions before and after in UCC-CVRM initiation.

### Guideline adherent cardiovascular risk factor assessment

Completeness of cardiovascular risk factor measurement ranged from 0% to 77% before UCC-CVRM (Fig 2; S1 Table). For all other risk factors, registration increased after UCC-CVRM initiation ranging from +11% (eGFR; from 76% to 87%) to +87% (physical activity; from 0% to 87%). For laboratory measurements, largest increase was seen for HbA1c

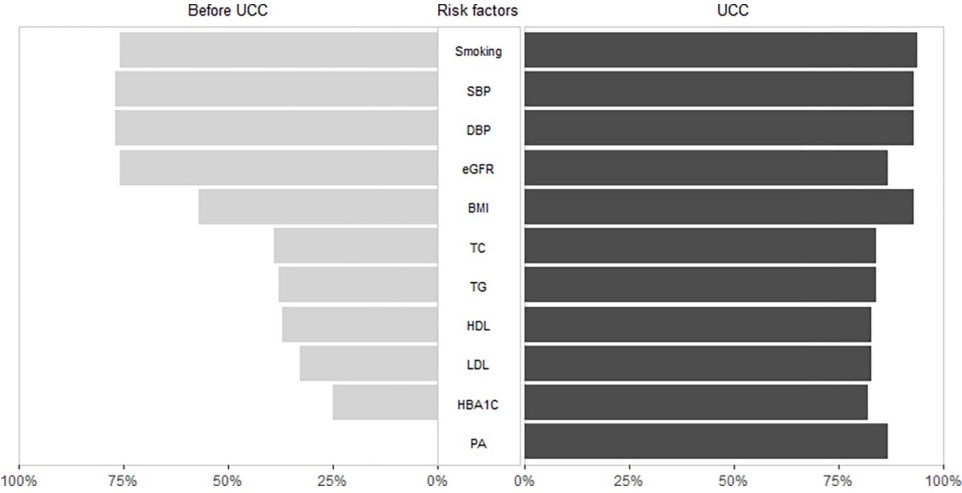

**Fig 2. Completeness of risk factor measurement before and after UCC-CVRM initiation.**

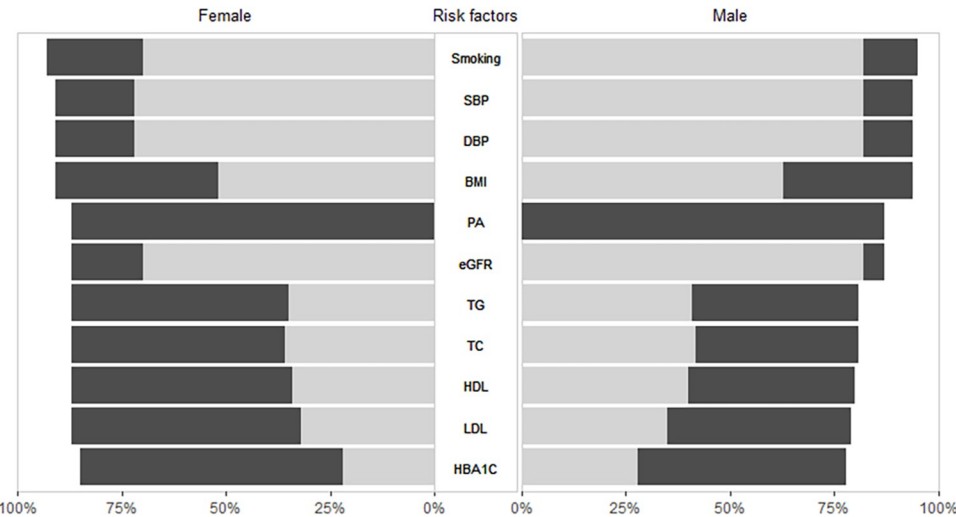

**Fig 3. Completeness of risk factor measurement before and after UCC-CVRM initiation stratified for sex.**

(from 25% to 82%, +57%). This leads to a completeness of all cardiovascular risk factor measurements ranging from 82% to 84% after UCC-CVRM initiation. Completeness of risk factor measurement was lower in women compared to men before UCC-CVRM for all risk factors (Fig 3, S2 and S3 Tables). After UCC-CVRM initiation, increase in measurement was higher in women, especially for lipids, eGFR and HbA1c, independent of age or co-occurrence of cardiovascular diseases and diabetes (Table 2). This means that the UCC-CVRM organization decreased the sex-gap in terms of a difference in measurement of risk factors in men and women.

## Detection of hypertension, dyslipidemia, and elevated HbA1c

Blood pressure was measured in 77% of patients before UCC-CVRM and in 93% of UCC-CVRM patients (S1 Table). Hypertension with an indication for treatment (change) was found in 45% of patients before UCC-CVRM and in 48% of UCC-CVRM patients. Combining this with the percentage of unmeasured patients (inverse of measured: 23% before

**Table 2. Factors associated with measurement of risk factors.**

| Factors | Measurement of risk factors | | | | |
|---|---|---|---|---|---|
| | Smoking | SBP | LDL-c | eGFR | HbA1c |
| *Intercept* | *0.98 (0.81–1.18)* | *0.31 (0.26–0.38)* | *0.53 (0.44–0.62)* | *0.62 (0.52–0.75)* | *0.30 (0.24–0.36)* |
| UCC-CVRM initiation | 4.26 (3.19–5.81) | 3.68 (2.79–4.94) | 6.99 (5.91–8.29) | 1.45 (1.18–1.80) | 8.73 (7.38–10.4) |
| Female sex | 0.63 (0.56–0.71) | 0.74 (0.66–0.84) | 0.86 (0.77–0.95) | 0.62 (0.55–0.70) | 0.75 (0.68–0.84) |
| UCC-CVRM initiation * Female sex [#] | 1.33 (0.89–1.97) | 1.35 (0.92–1.96) | 1.99 (1.53–2.59) | 2.11 (1.57–2.85) | 2.37 (1.83–3.08) |
| Age (year increase) | 1.02 (1.02–1.03) | 1.04 (1.04–1.04) | 1.00 (1.00–1.00) | 1.03 (1.03–1.03) | 1.00 (1.00–1.00) |
| CVD | 1.74 (1.50–2.01) | 2.94 (2.47–3.54) | 1.04 (0.93–1.15) | 1.86 (1.61–2.16) | 1.11 (0.99–1.24) |
| DM | 1.31 (1.09–1.59) | 1.29 (1.06–1.59) | 0.98 (0.85–1.13) | 1.54 (1.27–1.87) | 2.84 (2.46–3.28) |

UCC-CVRM–Utrecht Cardiovascular Cohort, SBP–Systolic Blood Pressure, LDL-C–Low-Density-Lipoprotein cholesterol, eGFR–estimated Glomerular Filtration Rate, HbA1c –glycated hemoglobin, CVD–history of a cardiovascular event including coronary heart disease/stroke/peripheral artery disease/abdominal aortic aneurysm, DM–history of diabetes mellitus.

[#]interaction of UCC-CVRM intervention and sex

UCC-CVRM and 7% after UCC-CVRM initiation), this leads to a likelihood to miss hypertension of $(0.23*0.45)*100\% = 10.4\%$ before UCC-CVRM and $(0.07*0.48)*100\% = 3.4\%$ after UCC-CVRM initiation. This translates to a risk of missing of $3.4/10.4 = 0.33$ after UCC-CVRM initiation: a 67% reduction of likelihood to miss uncontrolled hypertension after UCC-CVRM initiation (S3 Table). Similarly, LDL-c was measured in 33% of patients before UCC-CVRM and in 83% of UCC-CVRM patients. Dyslipidemia with an indication for treatment (change) was found in 65% of patients before UCC-CVRM and in 66% of UCC-CVRM patients. This translates to a 75% reduction of likelihood to miss uncontrolled dyslipidemia after UCC-CVRM initiation (S4 Table). Lastly, HbA1c was measured in 25% of patients before UCC-CVRM and in 82% of UCC-CVRM patients. Off target HbA1c with an indication for treatment (change) was found in 26% of patients before UCC-CVRM and in 11% of UCC-CVRM patients. This translates to a 90% reduction of likelihood to miss uncontrolled diabetes after UCC-CVRM initiation (S4 Table). The higher increase in measurement in women after UCC-CVRM initiation also translates to a larger reduction in likelihoods to miss uncontrolled hypertension, dyslipidemia, and diabetes in women (68%, 80%, and 93%, respectively) compared to men (61%, 70%, and 85%, respectively (all $p<0.0001$)).

## Time trends

Data collection depends on inclusion of patients in UCC-CVRM. Times with limited staff availability (holidays, flu period) proved to be challenging to keep invitation and informed consent rates constant. This is reflected in the mean number of inclusion consults per month: 50 for holiday months (May, August, September, and December) and 77 for non-holiday months in 2018. Although UCC-CVRM is phrased and positioned as a LHS and routine clinical care, in the developing phase it was not seen as routine care by supporting staff and was therefore not prioritized during clinic rush hours in times with limited staff.

## Feedback: Extreme values

A total of 233 extreme values were checked from 2016 to 2018 for guideline adherent management. From the clinical reports in the EHR, it was unclear if follow-up was provided according to the guidelines for 26 of the extreme values (11%). Of these 26, 11 were HbA1c values (42%) and 7 (27%) systolic blood pressure measurements and 8 other.

## Discussion

The development of a cardiovascular LHS through systematic registration of the cardiovascular risk profile in all patients referred for evaluation of an (a)symptomatic cardiovascular disease improves guideline adherence substantially and decreases risk of missing patients with elevated risk factors levels with an indication for treatment. This effect is even more pronounced in women compared to men, indicating that how and LHS approach can resolve selectivity in care and provide and inclusive view on health in the entire–in this case cardiovascular–patient population.

 The UCC-CVRM has set up a hospital-wide basis for a cardiovascular LHS starting with data collection in routine clinical care. The level of completeness of the cardiovascular risk assessment was significantly higher in UCC-CVRM compared to what we found before UCC-CVRM. Although guidelines prescribe evaluation of the cardiovascular risk profile in vascular patients referred to a hospital for further evaluation [14], the need for complete CVRM assessment might not have been recognized or prioritized considering this is a tertiary care center with high level complexity of diseases. The prevalence of abnormal HbA1c values was significantly higher before UCC-CVRM than after. Possibly, CVRM was checked by the

general practitioner and only high-risk patients were referred, or HbA1c was selectively measured in (per) diabetic patient, both results in an overestimation of prevalence of abnormal value and also missed uncontrolled HbA1c. For hypertension and dyslipidemia this, however, did not seem to hold. The UCC-CVRM approach resulted in a more inclusive population: especially in women the increase in measurement completeness was eminent. In both EURO-SPIRE and SWEDEHEART registration studies, women were underrepresented (24% and +/-26%, respectively). Women that were included were undertreated and failed to reach the treatment targets more frequently compared to men [3,4]. Amongst others, failure to recognize the importance of risk factor control and higher rates of reported drug side effects in women may have contributed to this difference [15,16].

When interpreting results on the effect of UCC-CVRM on the registration of risk factors one should take strengths and limitations into account. We proposed matching UCC-CVRM patients 1:4, which was impossible for some patients with outlier age (i.e. from the department of geriatrics) or with a very rare diagnosis. This might have resulted in a decrease of balance, yet we think this does not influence our results to the extent that it will change our main conclusions. Physical activity could not be retrieved for patients before UCC-CVRM. Within our EHR, there is no specific structured field defining physical activity nor do we have a text-mining algorithm to extract this from unstructured text. We therefore might have underestimated the registration of physical activity before UCC-CVRM and overestimated the impact of UCC-CVRM in this respect. Registration of smoking was high compared to other studies, which report approximately 68–94% availability of smoking status in structured EHR cells [17–19]. For our project, we used an in-house developed (free)text-mining algorithm tweaked to our EHR to retrieve smoking information, which is not restricted by structured fields or even specific parts in the EHR and therefore can extract more information from the EHR [20]. This smoking mining algorithm was validated on a UCC-CVRM set before and showed good diagnostic accuracy with exception of short-term current smoking because many patients had quit in the meantime resulting in a lower positive predictive value [19]. Potentially, similar overestimation of current smoking prevalence occurred for before UCC-CVRM patients, explaining the higher prevalence compared to UCC-CVRM. Another limitation of using routine care data for our analysis, as for any routine care data analyses, is the restriction to what is reported in the EHR. We assume here that everything what was measured, was reported in the EHR. As we used mostly lab-based or other very much structured data and a validated algorithm to text-mine smoking information [19], we think that our extractions are as complete as possible. Yet, not testing–thus reporting- a specific risk factor can also be regarded as meaningful as a) the factor was considered to be normal (for example: normal BMI appearance of a patient) and thus b) not relevant to clinical decision making (no proportion attributable risk by BMI for this patient). This missing not at random (MNAR, the choice for or against a reported factor contains meaning about this factor) pattern could have made us overestimate the likelihood of missing a clinically relevant risk factor [20]. Because these likelihoods are based on the assumption of equal distribution of these values among unreported values. Lastly, these data represent only patients visiting the outpatient clinic. UCC-CVRM organization for admitted patients has proven to be more challenging due to limited staff and burden for the patient as these are more diseased compared to outpatient visitors. We are investigating alternative routes to support complete cardiovascular risk factor assessment in these patients, potentially at their follow-up appointment after discharge.

Although the concept of LHS was known, an example of a practical application of it was lacking at the time UCC-CVRM was initiated [7]. We inevitably ran into challenges grown from pioneering. Since the UCC-CVRM was set up prospectively and also contains a biobank, interpretation of current regulation resulted in an informed consent requirement, which

might have influenced the inclusiveness of our population. An academic reflection on current ethical and legislative frameworks and how these are compatible with both the aims of a LHS and existing regulations whilst safeguarding patients' autonomy and wellbeing is now emerging [21,22]. Adoption of UCC-CVRM structure–most importantly the invitation and inclusion processes–required efforts to tailor these processes to department specific organizations and dedicated support from the Center of Circulatory Health Management. The paradigm shift of traditional healthcare system to an LHS requires a collaborative effort from all stakeholders, with strong leadership from stakeholder representatives [23].

UCC-CVRM has shown to promote optimal care, to provide feedback on the quality of care, and facilitate research. Now, sufficient quality and amount of data is available to support automated feedback processes and decision support, for these computerized decision support systems (CDSS) require complete input data [11]. A CDSS was developed to aid guideline adherent cardiovascular risk management [24]. With this tool, clinicians and patients are provided with an overview of their cardiovascular risk factors, a 10-years risk prediction for cardiovascular events, and guideline adherent suggestions for therapy [24]. CDSS can also be used for automated feedback on extreme values via alerts. On a higher level, CDSS for care evaluations, and benchmarking between clinicians or even hospitals. Formal cost-effectiveness evaluations regarding CDSS use for cardiovascular risk management are yet to be conducted.

## Conclusion

In a collaborative effort by care and research professionals, the UCC-CVRM has set a benchmark for a cardiovascular LHS. Systematic registration of the cardiovascular risk profile in all vascular patients referred for evaluation impacts tremendously on guideline adherence and risk of missing patients with elevated risk factors levels with an indication for treatment. An LHS approach provides insight into of quality of care within an more inclusive cross-section of cardiovascular population, ultimately leading to improved primary and secondary prevention of cardiovascular disease.

## Supporting information

**S1 Table. Cardiovascular risk factor measurement and distributions before and after UCC-CVRM initiation.**
(DOCX)

**S2 Table. Presence of risk factor measurement before and after UCC-CVRM initiation, stratified for sex.**
(DOCX)

**S3 Table. Risk factor distributions before and after UCC-CVRM initiation, stratified for sex.**
(DOCX)

**S4 Table. Likelihood to miss uncontrolled risk factor calculations.**
(DOCX)

## Acknowledgments

Members of the UPOD study group: Wouter van Solinge, Imo Hoefer, Saskia Haitjema, Mark de Groot. Members of the Utrecht Cardiovascular Cohort- CardioVascular Risk Management (UCC- CVRM) Study group: F.W. Asselbergs, Department of Cardiology; G.J. de Borst, Department of Vascular Surgery; M.L. Bots (chair),Julius Center for Health Sciences and

Primary Care; S. Dieleman, Division of Vital Functions (anesthesiology and intensive care); M.H. Emmelot, Department of Geriatrics; P.A. de Jong, Department of Radiology; A.T. Lely, Department of Obstetrics/Gynecology; I.E. Hoefer, Laboratory of Clinical Chemistry and Hematology; N.P. van der Kaaij, Department of Cardiothoracic Surgery; Y.M. Ruigrok, Department of Neurology; M.C. Verhaar, Department of Nephrology & Hypertension, F.L.J. Visseren, Department of Vascular Medicine, University Medical Center Utrecht and Utrecht University.

## Author Contributions

**Conceptualization:** Diederick E. Grobbee, Folkert W. Asselbergs, Michiel L. Bots.

**Data curation:** T. Katrien J. Groenhof, Saskia Haitjema, A. Titia Lely, Folkert W. Asselbergs, Michiel L. Bots.

**Formal analysis:** T. Katrien J. Groenhof, Saskia Haitjema, Michiel L. Bots.

**Funding acquisition:** Diederick E. Grobbee, Folkert W. Asselbergs, Michiel L. Bots.

**Methodology:** Saskia Haitjema, A. Titia Lely, Diederick E. Grobbee, Folkert W. Asselbergs, Michiel L. Bots.

**Resources:** Diederick E. Grobbee.

**Supervision:** Saskia Haitjema, A. Titia Lely, Folkert W. Asselbergs, Michiel L. Bots.

**Visualization:** T. Katrien J. Groenhof.

**Writing – original draft:** T. Katrien J. Groenhof.

**Writing – review & editing:** Saskia Haitjema, A. Titia Lely, Diederick E. Grobbee, Folkert W. Asselbergs, Michiel L. Bots.

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
