## [Decision Letter · Decision Letter 0]

14 Nov 2022

PDIG-D-22-00238

Optimizing cardiovascular risk assessment and registration in a developing cardiovascular learning health care system: women benefit most

PLOS Digital Health

Dear Dr. Bots,

Thank you for submitting your manuscript to PLOS Digital Health. After careful consideration, we feel that it has merit but does not fully meet PLOS Digital Health's publication criteria as it currently stands. Therefore, we invite you to submit a revised version of the manuscript that addresses the points raised during the review process.

Please submit your revised manuscript within 30 days Dec 14 2022 11:59PM. If you will need more time than this to complete your revisions, please reply to this message or contact the journal office at digitalhealth@plos.org. Please include the following items when submitting your revised manuscript:

We look forward to receiving your revised manuscript.

Kind regards,

Alistair Johnson

Section Editor

PLOS Digital Health

Journal Requirements:

a State the initials, alongside each funding source, of each author to receive each grant.

3. Please provide separate figure files in .tif or .eps format only and remove any figures embedded in your manuscript file. Please also ensure that all files are under our size limit of 10MB.

4. We noticed that you used "data not shown" in the manuscript. We do not allow these references, as the PLOS data access policy requires that all data be either published with the manuscript or made available in a publicly accessible database. Please amend the supplementary material to include the referenced data or remove the references.

5. We notice that your supplementary tables are included in the manuscript file. Please remove them and upload them with the file type 'Supporting Information'. Please ensure that each Supporting Information file has a legend listed in the manuscript after the references list.

Additional Editor Comments (if provided):

Apologies for the delay in reviewing this manuscript. Upon receipt of the review I am recommending minor revision.

Reviewers' comments:

Reviewer's Responses to Questions

**Comments to the Author**

1. Does this manuscript meet PLOS Digital Health’s publication criteria? Is the manuscript technically sound, and do the data support the conclusions? The manuscript must describe methodologically and ethically rigorous research with conclusions that are appropriately drawn based on the data presented.

Reviewer #1: Yes

2. Has the statistical analysis been performed appropriately and rigorously?

Reviewer #1: Yes

3. Have the authors made all data underlying the findings in their manuscript fully available (please refer to the Data Availability Statement at the start of the manuscript PDF file)?

Reviewer #1: No

4. Is the manuscript presented in an intelligible fashion and written in standard English?

Reviewer #1: Yes

5. Review Comments to the Author

Reviewer #1: This is an interesting manuscript that demonstrates a real-world application of the learning healthcare system (LHS) applied in the context of cardiovascular risk assessment and management. Grohenhof et al. clearly state the problem definition. Notably, they present LHS as a solution to what they define as the science-to-care gap. 

The authors conducted a prospective cohort study comparing the measurement of cardiovascular risk factors before and after the implementation of the UCC-CVRM, an infrastructure implemented in the EHR allowing for standardized collection of cardiovascular-related health data. To demonstrate the benefits of this infrastructure, the authors compared the proportion of measured risk factors before and after its implementation. The other outcomes measured by the authors were the proportion of patients with an indication of treatment for a given cardiovascular condition and an estimation of the proportion of patients with potentially missed severe cardiovascular-related conditions.

The results are interesting as they have shown an increase in the registration of all risk factors after implementing UCC-CVRM (see S1 Table and S2 Table). The authors extrapolate their results a little bit further by estimating the likelihood of missing uncontrolled risk factor in patients who would have otherwise required treatment (S4 Table).

I have the following comments :

Major issues

Abstract and Introduction

1) No explicit mention of similar LHS initiatives in the cardiovascular space. Notably, there seem to have other examples in the literature that have implemented LHS in this space, and it would be essential to cite and explain how your initiative differs. In the cited article of Maddox et al., multiple examples are discussed. 

Methods

2) Please add p-values to show the significance of the multiple statistical tests conducted (S1 Table, S2 Table

3) I don’t agree with the calculation of the likelihood of missing hypertension: in the context of EHR documentation, not being reported does not equate not being measured. Moreover, your calculation implies a uniform distribution among unmeasured variables, which is also incorrect. I like the idea you explored with those calculations, but I would like to see a few sentenced conceding the limitation of the methodology. 

Figures

4) Figure 5 and 6 do not allow for easy comparison of the pre/post CVRM implementation. I would suggest using side-by-side bar chart or use the S1 and S2 tables instead. 

Minor issues

1) Minor grammar issues and sentence structures could benefit from some touch-up.

i.e. lines 48-50 could be rewritten for simplicity «Healthcare is challenged by a growth in patient numbers, patients of higher age, patients with multiple diseases and medications, and a higher level of chronicity of disease. » 

2) Clarification of the statement line 275. Please clarify how CDSS relates or complements the work you presented in this manuscript. The last paragraph was not so clear to me. 

3) Add a discussion regarding the interaction of CVRM initiation and female sex. Notably, elaborate on the potential reasons for which the results are significant for eGFR, A1c and not for SBP and smoking.

6. PLOS authors have the option to publish the peer review history of their article (what does this mean?). If published, this will include your full peer review and any attached files.

**Do you want your identity to be public for this peer review?** For information about this choice, including consent withdrawal, please see our Privacy Policy.

Reviewer #1: Yes: Eric Yamga

---

## [Editor Report · Decision Letter 1]

30 Dec 2022

Optimizing cardiovascular risk assessment and registration in a developing cardiovascular learning health care system: women benefit most

PDIG-D-22-00238R1

Dear prof Bots,

We are pleased to inform you that your manuscript 'Optimizing cardiovascular risk assessment and registration in a developing cardiovascular learning health care system: women benefit most' has been provisionally accepted for publication in PLOS Digital Health.

Best regards,

Leo Anthony Celi, MD MS MPH

Editor-In-Chief

PLOS Digital Health